# Evaluation of Meningococcal Serogroup C Bactericidal Antibodies after Primary Vaccination: A Multicentre Study, Italy

**DOI:** 10.3390/vaccines10050778

**Published:** 2022-05-14

**Authors:** Arianna Neri, Massimo Fabiani, Anna Maria Barbui, Caterina Vocale, Alessandro Miglietta, Cecilia Fazio, Anna Carannante, Annapina Palmieri, Paola Vacca, Luigina Ambrosio, Paola Stefanelli

**Affiliations:** 1Department Infectious Diseases, Istituto Superiore di Sanità, 00161 Rome, Italy; arianna.neri@iss.it (A.N.); massimo.fabiani@iss.it (M.F.); cecilia.fazio@iss.it (C.F.); anna.carannante@iss.it (A.C.); paola.vacca@iss.it (P.V.); luigina.ambrosio@iss.it (L.A.); 2Microbiology and Virology Laboratory, Città della Salute e della Scienza Hospital, 10126 Torino, Italy; abarbui@cittadellasalute.to.it; 3Microbiology Unit, IRCCS Azienda Ospedaliero-Universitaria di Bologna, 40138 Bologna, Italy; cate.vocale@gmail.com; 4Central Tuscany Health Authority, 50142 Florence, Italy; alessandro.miglietta@uslcentro.toscana.it; 5Department of Cardiovascular, Endocrine-Metabolic Diseases and Aging, Istituto Superiore di Sanità, 00161 Rome, Italy; annapina.palmieri@iss.it

**Keywords:** bactericidal antibody, *Neisseria meningitidis*, meningococcal conjugate vaccine

## Abstract

Here, we evaluated over time in different cohorts of children vaccinated against serogroup C *Neisseria meningitidis*, the presence of antibodies with neutralizing activity. A total of 348 sera samples of enrolled children by year since vaccination (<1 year- up to 5 years), starting from February 2016 to December 2017, were collected in three collaborating centers. Meningococcal serogroup C (MenC) antibody titers were measured with a serum bactericidal antibody (SBA) assay using rabbit complement (rSBA) following standard operating procedures. The cut-off of rSBA titer ≥ 8 is considered the correlate of protection. We observed a significantly declining of bactericidal rSBA titers by 23% every year, for every 1-year from vaccination (Adjusted PR = 0.77, 95% CI: 0.71–0.84). The proportions of children with bactericidal antibodies, immunized with the meningococcal serogroup C conjugate (MCC) vaccine, declined from 67.7% (95% CI: 48.6–83.3%) one year after vaccination, to 36.7% (95% CI: 19.9–56.1%) five years after vaccination (chi-square for linear trend, *p* < 0.001). Children vaccinated with the tetravalent meningococcal serogroup ACWY vaccine resulted in a high proportion of bactericidal rSBA MenC titer ≥ 1:8 (90.6%, 95% CI: 79.3–96.9%) after a mean time of seven months. Overall, the results provide some evidences on the evaluation of meningococcal serogroup C bactericidal antibodies after primary vaccination.

## 1. Introduction

In Italy the annual incidence rate of invasive meningococcal disease (IMD) declined from 0.6 per 100,000 in 2004 to 0.3 in 2007 [1] remaining low until 2019 [2]. This was mainly due to a decrease in meningococcal serogroup C (MenC) cases after the introduction of the MenC conjugate (MCC) vaccine, which was included in the 2005–2007 National Plan for Vaccine Prevention (PNPV) [3]. The MCC vaccine was recommended to infants in the first year of life and to all individuals presenting splenic dysfunction or immunodeficiency [3]. As a consequence, a significant impact on MenC disease in infants and children (aged 0 to 4 years) was observed. MenC disease decreased from 1.7 per 100,000 in 2004 to 0.5 per 100,000 in 2007 [1] and was 0.17 per 100,000 in 2019 in the age group 0–4 years [2].

In 2008, the MCC vaccine was recommended also to adolescents aged 11–18 years old within the 2012–2014 PNPV [4]. In 2012, the tetravalent meningococcal serogroup ACWY conjugate (MenACWY) vaccine was introduced and in the 2017–2019 PNPV it was recommended for individuals (aged from 12 to 18 years) and not previously vaccinated and for children already immunized [5].

Since their introduction in Italy, the vaccine coverage rate for MCC and MenACWY vaccines increased up to 2018: the vaccine coverage rate was highest in children aged 24–36 months for the MCC vaccine and in adolescents aged 16 years for the MenACWY vaccine [6]. In 2019–2020, MCC vaccine coverage decreased while MenACWY vaccination rates tended to increase [7].

The protective effect against the capsular polysaccharide C included in the MCC vaccine is associated with the decline in the number of MenC cases in the targeted age groups in different countries. In the United Kingdom (UK), the MCC vaccine introduction at the end of 1999 in children and young adults markedly reduced the incidence of MenC disease, in the first two years [8]. Since then, other European countries, Canada and Australia, who also introduced the MCC vaccine, have reported a substantial decline in MenC disease incidence [9,10,11].

Although many clinical studies showed high effectiveness and immune memory of different meningococcal conjugate vaccines, a waning of bactericidal antibodies following primary vaccination in young children was evidenced [12,13,14]. For example, two studies in the United Kingdom reported a decrease of SBA titers after 3 doses of MCC vaccine during the first year of life and after a single dose at 1–3 years of age of MCC vaccine two years after vaccination [15,16].

Other studies have shown that protection against MenC disease provided by quadrivalent conjugate vaccines received in toddlers and children may decline over time [17,18,19].

Since antibody persistence is important as a parameter to monitor protection against meningococcal disease, we conducted a study to assess serum bactericidal antibody (SBA) persistence against serogroup C polysaccharide in samples of children who received primary vaccination with MCC or MenACWY vaccines.

## 2. Methods

This study was conducted at the National Reference Laboratory (NRL) for IMD at Istituto Superiore di Sanità (ISS) in collaboration with three centers: St. Orsola-Malpighi Polyclinic, in Bologna, Molinette Hospital, in Turin and Central Tuscany Local Health Authority, in Florence. The protocol including recruitment and serum sampling of MenC vaccinated individuals, was drawn up by the NRL. The enrolment of participating was from February 2016 to December 2017. Serum samples were collected from healthy children who belonged to birth cohort from 2004 to 2016 and who had received the primary MenC vaccination, with the MCC or MenACYW vaccine. These children were recruited after the approval of all parents who responded to the letter sent to them. All participants were divided into two groups: children who received the MCC vaccine (MCC-CRM_197_), one or three doses (3, 5 and >12 months), or children who received one dose of the MenACYW vaccine (MenACYW-TT and MenACYW-CRM_197_). An additional group enclosing children who received a booster dose with the MCC vaccine or MenACYW vaccine given 4 to 10 years after primary MCC vaccination (one or three doses of MCC vaccine) was analyzed separately to other groups of enrolled individuals. Exclusion criteria included immunosuppression and any acute or chronic infection.

Information on the date of birth, date of sample collection, and date of vaccination of the children was available. Written informed consent was obtained from parents or legally authorized representatives prior to the performance of any procedures.

All serum samples collected from each individual were sent to the NRL for IMD of the ISS and stored at −80 °C. Serum samples were taken from <1 year up to 5 years after vaccination, to determine the persistence of specific bactericidal antibody against serogroup C meningococcus.

### 2.1. Ethical Committee Approval

This study was approved by the ISS Ethical Committee and by the Local EC (ISS EC reference number PRE 926/15).

### 2.2. Serological Assay

A standardized SBA assay was performed according to the procedure described by Maslanka et al. [20], using baby rabbit complement (rSBA), as the source of exogenous complement. The complement source was a pooled serum from 3 or 4-week-old–rabbits (Cederlaine Corporation, 4410 Paletta Court, Burlington, Ontario, Canada). Titers were expressed as the reciprocal serum dilutions yielding ≥ 50% killing after 60 min. The titer of ≥1:8 was the protective threshold, that has been established to be the correlated of protection for meningococcal serogroup C [21]. The lower limit of detection was a titer of 4. Negative results were given a value of half the lowest level of detection. rSBA assays were performed against the serogroup C target strain C11 (C:16: P1.7-1,1). A positive control serum with an assigned serogroup C titer was included to quality control the assay.

### 2.3. Statistical Analysis

We described the main characteristics of the study sample using counts with percentages and range values. Based on date of birth, date of vaccination, and date of serum sampling we calculated the age at vaccination, the age at serum sampling, and the time interval from vaccination to serum sampling. We then estimated the proportion of children with a bactericidal antibody response (rSBA MenC titers ≥ 1:8), together with a 95% exact binomial confidence interval (CI), by age at serum sampling and different time intervals from serum sampling to vaccination, separately for each vaccine type (MCC and MenACYW). Changes in the proportion of children with bactericidal antibody response over the age at serum sampling and time from vaccination were evaluated through the chi-square test for linear trend.

Finally, separately for each vaccine type, we used a multivariable Poisson regression model with robust variance estimator to evaluate the association of bactericidal antibody response with age at vaccination and time from vaccination. The latter variables were included as continuous covariates in the model and the strength of the associations was described using adjusted prevalence proportion ratios (PPR) with 95% CI. All tests were two-sided and statistical significance was set at *p* < 0.05. The analyses were performed using Stata/SE version 16.1 (StataCorp LLC, Lakeway Dr, College Station, TX, USA).

## 3. Results

The study sample consisted of 348 enrolled participants, 272 of whom were included in the analysis. The exclusion from the analysis was due to serological data missing or to blood samples drawn before MenC immunization.

The age at the time of blood sampling of all investigated children ranged from 1 year to 12 years. These subjects were grouped based on the type of MenC vaccine administered: 203 (74.6%) had received MCC vaccine (one dose or three doses), 53 (19.5%) had received one dose of MenACYW vaccine and 16 children (5.9%) had received a booster dose with MCC or MenACYW vaccine.

The age at MCC vaccination was between 7 months and 23 months for the 93.1% (189/203) and only four children (2%; 4/203) were vaccinated at the age of 5–11 years (Table 1). Children vaccinated with one dose of the MCC vaccine were the majority (94%; 191/203), while the remaining children received three doses (6%; 12/203).

The percentages of children in the MCC vaccine group with rSBA MenC titers ≥ 1:8 are presented in Figure 1 according to the age at enrollment. The highest percentages with bactericidal titers resulted in children under two years of age (90.9%, 95% CI: 70.8–98.9%) and at two years of age (70.4%, 95% CI: 49.8–86.2%). Another high percentage of children with bactericidal titers was found in the age group 7 to 12 years at enrollment (88.9%, 95% CI: 51.8–99.7%), vaccinated at the median age of 5.0 years (from 2.1 to 5.5 years). While children aged 3, 4, 5 and 6 years at enrollment showed the lowest percentages of bactericidal titers: 50% (95% CI: 31.3–68.7%), 46.2% (95% CI: 26.6–66.7%), 25.5% (95% CI: 14.7–39.0%), and 23.5% (95% CI: 10.7–41.2%), respectively (Figure 1). This decline (from 3 to 6 years of age at enrollment) was found to be statistically significant (chi-square for trend, *p* = 0.006).

In order to assess bactericidal antibody persistence over time, we analyzed the serum samples according to time since MCC vaccination. Figure 2 shows a decreasing trend of antibody levels of rSBA MenC titres ≥ 1:8 from 67.7% (95% CI: 48.6–83.3%) at one year after vaccination to 36.7% (95% CI: 19.9–56.1%) at five years after vaccination (chi-square for linear trend, *p* < 0.001).

Selecting only children who have been vaccinated for more than four years with the MCC vaccine, the proportion with rSBA MenC titers ≥ 1:8 was higher in children vaccinated at more than 5 years old (75%) than in those who received the vaccine at under 2 years old (29.7%). This difference was not significant due to the little number of children immunized at >5 years old (data not shown).

As showed in Table 1 the percentage of the children who received one dose of the MenACYW vaccine was 84.9%, between 1 to 2 years of age and only three children were vaccinated between 5 and 7 years. Among these subjects, the majority, showed a high percentage of rSBA MenC titers ≥ 1:8 (90.6%, 95% CI: 79.3–96.9%) after a mean time of 7 months (range 1 month–17 months).

We finally evaluated the percentage of rSBA MenC titers ≥ 1:8 in a small group (n = 16) of children who received a booster dose of MCC or MenACYW vaccine 4 to 10 years after primary MCC vaccination, performed at one year. All the children showed rSBA MenC titers ≥1:256 in a mean time of 4 months (range 1 month–13 months) after the booster dose (data not shown).

According to results from the multivariable Poisson regression analysis model presented in Table 2, independently of the age at vaccination, the proportion of children vaccinated with the MCC vaccine with a bactericidal antibody response (rSBA titers ≥1:8) significantly decreased by 23% for every 1-year increase in time from vaccination (Adjusted PR = 0.77, 95% CI:0.71–0.84). Conversely, independently of the time since vaccination, it significantly increased by 10% for every 1-year unit increase in age at vaccination (Adjusted PR = 1.10, 95% CI: 1.01–1.20). No statistically significant trends were observed among children vaccinated with the MenACYW vaccine.

## 4. Discussion

Meningococcal disease is still a devastating threat characterized by high fatality rates due to the rapid progression of the disease [22]. An important advance against this public health problem was the development and introduction of polysaccharide-protein conjugate vaccines. In particular, the introduction of the monovalent serogroup C meningococcal conjugate vaccine had great success in the direct protection against meningococcal disease [23].

It is known that protection against meningococcal disease is mediated by the presence of serum bactericidal antibodies induced by meningococcal polysaccharide protein conjugates vaccines [24]. The capsule polysaccharide is the major virulence determinant that allows the meningococcus to survive in the blood and in the intracellular environment [25]. In addition, meningococcal capsule polysaccharide can interfere with the complement activation of both classical and alternative pathways [26]. The complement system has an important role in protection against meningococcal disease, evidenced by the increased susceptibility of individuals with known terminal complement deficiencies [27].

The rSBA assay was used in this study to measure the levels of the bactericidal antibodies for MenC in a sample of children who received primary vaccination with the MCC and MenACWY vaccine.

At the time of our study, meningococcal vaccination against serogroup C was recommended for all children aged 13–15 months with a single dose of MCC or MenACWY vaccine [28]. Consequently, mostly of the children enrolled in this study received a single dose of the MCC vaccine between 7 and 23 months of age. As expected, the high proportion of samples with a rSBA value ≥ 1:8 was found among children under two years of age at the time of sampling, followed by children aged between 7 and 12 years. Thus was probably due to the age of immunization because these older children received a single dose of the MenC vaccine at a median age of 5 years. Previous data underlined how the SBA protective titers against serogroup C meningococci, after the MCC vaccine, increase with increasing age at the time of vaccination [29].

Considering that the majority of children in this study were vaccinated at one or two years of age and that they belonged to different birth cohorts (from 2004 to 2016), we analyzed the serum samples in relation to the time of vaccination. We found that bactericidal antibodies declined over time from 89.3% at one year, to 36.7% 5 years after MCC vaccination. This percentage is consistent with previous studies that demonstrated marked waning in rSBA MenC titers at 2 and 4 years after MCC immunization of children at a younger age [16,30].

Long-term protection after a single dose of the MenACWY vaccine has been evidenced in children and adolescents and revealed the serogroup-specific decline in titers, from 2 years up to 10 years [31,32,33,34]. In this study, the high proportion of protective rSBA MenC titers was found in a group of children immunized with the MenACYW vaccine, approximately at a mean time of 7 months after vaccination. This result is in line with what was reported in a previous study in which most children showed a rSBA MenC ≥1:128 after 1-month post-vaccination with the MenACYW vaccine [35].

The evaluation of the persistence of bactericidal antibodies in the serum, it is important not only because is associated with protection but also to provide evidence for vaccination programs and to determine the need of booster dose, required after primary vaccination [19]. The immune response to booster dose with MenACWY vaccine has been evaluated in toddlers, children, and adolescents as already reported [23].

Here, it was evaluated the presence of rSBA titers against serogroup C in a small group of children who received a MenACWY booster dose from 4 to 10 years after primary vaccination with MCC. This group showed a robust response due to the key role of the immunological memory [31].

In conclusion, our findings provide additional evidences on the presence of bactericidal antibodies after MCC or MenACWY vaccination. As already known, meningococcal bactericidal antibodies against serogroup C were observed to decline over time after infant and child MenC immunization. A significant proportion of children, vaccinated at 1–2 years showed low bactericidal antibody levels five years after MenC immunization. The impact of MenC disease on different age groups should be closely monitored to evaluate whether it is necessary to recommend booster doses of the meningococcal conjugate vaccine to maintain elevated bactericidal antibody levels.

## Figures and Tables

**Figure 1 vaccines-10-00778-f001:**
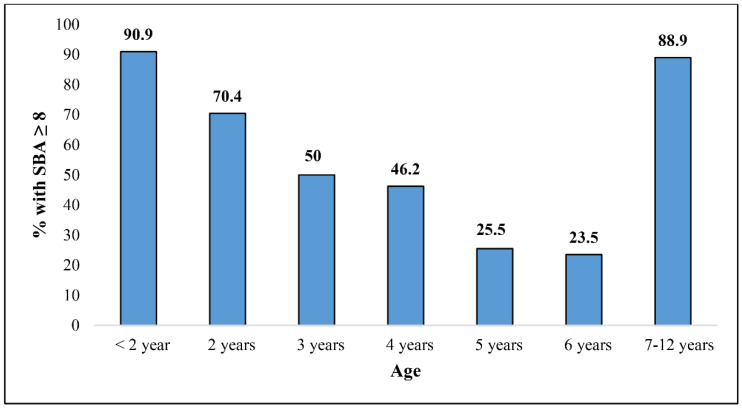
Percentage of children with MenC rSBA titers ≥ 8 by age at enrollment post MCC vaccine.

**Figure 2 vaccines-10-00778-f002:**
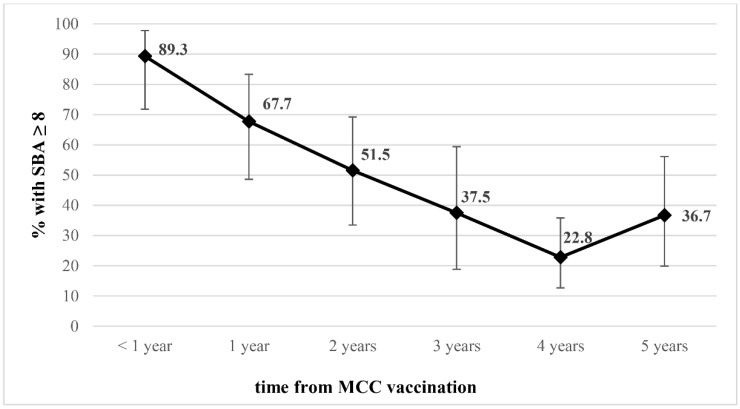
Percentage of children with serogroup C rSBA titers of ≥1:8 over time after MCC vaccination.

**Table 1 vaccines-10-00778-t001:** Number of enrolled participants distributed by age at vaccination and type of vaccine.

Age at Vaccination	MCC	ACYW	Booster Vaccination ^a^	Total
	No. Vaccinated (%)			
<2 years (7–23 months)	189 (93.1)	45 (84.9)	0 (0.0)	234 (85.1)
2 years (24–35 months)	7 (3.5)	1 (1.9)	0 (0.0)	8 (2.9)
≥3 years (36–133 months)	7 (3.5)	7 (13.2)	16 (100.0)	30 (12.0)
Total	203 (100.0)	53 (100.0)	16 (100.0)	272 (100.0)

**^a^** Booster vaccination with MCC or MenACWY vaccine.

**Table 2 vaccines-10-00778-t002:** Multivariable analysis of factors affecting protective response (rSBA titres ≥ 1:8) after MCC vaccination.

MCC Vaccine			
	Adjusted PPR	95% CI	*p*-Value
Time since vaccination	0.77	0.71–0.84	<0.001
Age at vaccination	1.10	1.01–1.20	0.024
**ACWY Vaccine**			
Time since vaccination	0.87	0.71–1.06	0.171
Age at vaccination	1.03	0.99–1.06	0.145

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
