# Peer review of "Evaluation of Meningococcal Serogroup C Bactericidal Antibodies after Primary Vaccination: A Multicentre Study, Italy"

_vaccines, 2022, doi:10.3390/vaccines10050778_

Round 1

Reviewer 1 Report

This short manuscript by Neri et al explores the waning over time of protective antibodies against MenC in children following immunization with MCC or MenACWY vaccines. The authors use standard bactericidal assays (utilizing rabbit complement) to demonstrate that the proportion of vaccinees with SBA titres ≥8 decreases over time, with a clear majority of vaccinees seemingly unprotected five years post-vaccination. The data presented is in agreement with several other studies (notably from the UK) which reveal similar findings. The study is well designed and technically sound, but the submission could be improved by changes to the text. The comments below are mostly regarding improvements to the English language, which would improve clarity.

Minor comments:

Line 20 – “seroprotective” should be “seroprotected”

Lines 24-27 – this needs rewording for clarity. Suggest “On the basis of rSBA titers, a significant proportion of children were not considered to be protected from MenC five years post-MCC vaccination. Overall the results provide evidence for the need for booster doses of vaccine against MenC to maintain a high level of antibodies capable of neutralizing the bacterium”.

Line 39 – should read “a significant impact on MenC disease in infants and children (aged 0 to 4 years) was observed”.

Lines 57-58 – “Since then, other European countries, Canada and Australia, who also introduced the MCC vaccine, have reported….”

Line 83 – “and” should be “or”.

Line 91 – “store” should be “stored”.

Line 100 – “rabbit” should be “rabbits”.

Line 176 – “partecipants” should be “participants”.

Table 1 – “2 years (24-34 months)”, should this be “2 years (24-35 months)”?

Table 1 – row total “31” – should this be “30”? Please check the figures in Table 1 throughout.

Line 187 – “is” should be “was”.

Line 232 – “is” should be “it is”.

Author Response

Revision of manuscript ID: vaccines-1714955 R1

Evaluation Of Meningococcal Serogroup C Bactericidal Antibodies After Primary Vaccination: A Multicentre Study, Italy

We thank the reviewer for their helpful revision of the manuscript. The referee’s comments have been carefully taken into account and the suggested text modifications have been introduced in this revised version of the manuscript. We have also considered some modifications to the text, in the Discussion section and the References.

Response to Reviewer 1 Comments

Line 20 – “seroprotective” should be “seroprotected”

  • Ok, we changed accordingly .

Lines 24-27 – this needs rewording for clarity. Suggest “On the basis of rSBA titers, a significant proportion of children were not considered to be protected from MenC five years post-MCC vaccination. Overall the results provide evidence for the need for booster doses of vaccine against MenC to maintain a high level of antibodies capable of neutralizing the bacterium”.

  • Ok, we changed accordingly.

Line 39 – should read “a significant impact on MenC disease in infants and children (aged 0 to 4 years) was observed”.

  • Ok, we changed accordingly.

Lines 57-58 – “Since then, other European countries, Canada and Australia, who also introduced the MCC vaccine, have reported….”

  • Ok, we changed as suggested.

Line 83 – “and” should be “or”.

  • Ok, we changed as suggested.

Line 91 – “store” should be “stored”.

  •  Ok

Line 100 – “rabbit” should be “rabbits”.

  •  Ok

Line 176 – “partecipants” should be “participants”.

  • Ok, we changed accordingly.

Table 1 – “2 years (24-34 months)”, should this be “2 years (24-35 months)”?

  • Yes, we changed.

Table 1 – row total “31” – should this be “30”? Please check the figures in Table 1 throughout.

  • Yes, it is 30. We corrected.

Line 187 – “is” should be “was”.

  •  Ok

Line 232 – “is” should be “it is”.

  •  Ok

We hope that our revisions adequately address the reviewer’ points. Please, let us know if there are further questions or requests.

Kind regards,

Paola Stefanelli on behalf of all co-authors

Dept. Infectious Diseases,

Istituto Superiore di Sanità, Rome, Italy

Reviewer 2 Report

The manuscript needs some language corrections before being published. English language editing by a native English speaker would allow easier and pleasant reading of the manuscript.

The presentation of the results is not clear and the figures and tables are not well designed.

The conclusions are not clear and do not reflect the findings obtained in the work.

Specific comments: Please define all the abbreviations the first time they are used (e.g. MCC, Men ACYW, SBA, ISS and others).

Author Response

Revision of manuscript ID: vaccines-1714955 R2

Evaluation Of Meningococcal Serogroup C Bactericidal Antibodies After Primary Vaccination: A Multicentre Study, Italy

We thank the reviewer for their helpful revision of the manuscript. The referee’s comments have been carefully taken into account and the suggested text modifications have been introduced in this revised version of the manuscript. We have also considered some modifications to the text, in the Discussion section and the References.

Response to Reviewer 2 Comments

The manuscript needs some language corrections before being published. English language editing by a native English speaker would allow easier and pleasant reading of the manuscript.

  • As required, the English has been considered through the new version.

The presentation of the results is not clear and the figures and tables are not well designed.

  • We do agree and the results, the figures and tables were improved.

The conclusions are not clear and do not reflect the findings obtained in the work.

  • We agree with the reviewer we implemented accordingly.

Specific comments: Please define all the abbreviations the first time they are used (e.g. MCC, Men ACYW, SBA, ISS and others).

  • Ok, we have defined all the abbreviations used for the first time in the manuscript accordly.

We hope that our revisions adequately address the reviewer’ points. Please, let us know if there are further questions or requests.

Kind regards,

Paola Stefanelli on behalf of all co-authors

Dept. Infectious Diseases,

Istituto Superiore di Sanità, Rome, Italy

Reviewer 3 Report

the paper from Neri et al is a straightforward study of the levels of SBA in childern vaccinated with MenC vaccines in Italy. The only limitation of the study the authors should note is the lowish number of sera tested for this duration of time studied. I have only a few minor comments/questions/suggestions

  1. English language in general is fine, though needs to be checked in some places.
  2. Avoid using webpages in introduction. Better to include them as true citations. 
  3. SBA assay needs bit more details - is there a citation for using strain C11 (why is this used?). Also, what is the positivie control serum for the assay - single human or pooled serum with high SBA or an animal serum?
  4. Results are fine.
  5. Discussion - in general fine, though I have one question that the authors must comment on. Was there any reported incidence of MenC disease amongst any of the children in the study despite declining SBA titres over time?
  6. Can it be clarified - did individual children from the point of vaccination have serum samples taken at yearly intervals for 5 years to test their SBA? 
  7. Comment needed from authors - even if SBA decline (and we know it is a correlate for protection), does this mean that the children will lose protection because they lose immunological memory? I think the conclusion line 245- 246 should be changed and to say that they are 'not protected' is wrong without providing evidence of disease amongst this group and an abscence of immunological memory cells.

Author Response

Revision of manuscript ID: vaccines-1714955 R3

Evaluation Of Meningococcal Serogroup C Bactericidal Antibodies After Primary Vaccination: A Multicentre Study, Italy

We thank the reviewer for their helpful revision of the manuscript. The referee’s comments have been carefully taken into account and the suggested text modifications have been introduced in this revised version of the manuscript. We have also considered some modifications to the text, in the Discussion section and the References.

Response to Reviewer 3 Comments

  1. English language in general is fine, though needs to be checked in some places.
  • As required, the English has been considered through the new version.

  1. Avoid using webpages in introduction. Better to include them as true citations. 
  • Ok, we have modified webpages in introduction as true citations, as indicated. The Reference list has been modified.

  1. SBA assay needs bit more details - is there a citation for using strain C11 (why is this used?).
  • Yes, the strain C11 is mentioned in the reference 20. This strain is used in the SBA assay because during the standardization of Neisseria meningitidis serogroup C Serum Bactericidal Antibody assay the strain C11 was established to be the reference target strain.

Also, what is the positivie control serum for the assay - single human or pooled serum with high SBA or an animal serum?

  • The positive control used in our study is a pooled serum from adults with an assigned high serogroup C SBA titer.

  1. Results are fine.
  • Ok, thank you.

  1. Discussion - in general fine, though I have one question that the authors must comment on. Was there any reported incidence of MenC disease amongst any of the children in the study despite declining SBA titres over time?
  • Unfortunately, we have not information about the occurrence of MenC disease in children enrolled in this study.

  1. Can it be clarified - did individual children from the point of vaccination have serum samples taken at yearly intervals for 5 years to test their SBA?
  • No, from a sample of 348 sera we selected the samples based on their distance from the date of MenC vaccination.

  1. Comment needed from authors - even if SBA decline (and we know it is a correlate for protection), does this mean that the children will lose protection because they lose immunological memory?
  • The SBA assay measures functional antibody levels, but not the presence of immunological memory.

  1. I think the conclusion line 245- 246 should be changed and to say that they are 'not protected' is wrong without providing evidence of disease amongst this group and an abscence of immunological memory cells.
  • We do agree and we implemented accordingly.

We hope that our revisions adequately address the reviewer’ points. Please, let us know if there are further questions or requests.

Kind regards,

Paola Stefanelli on behalf of all co-authors

Dept. Infectious Diseases,

Istituto Superiore di Sanità, Rome, Italy